# Carcinoembryonic Antigen Expression in Human Tumors: A Tissue Microarray Study on 13,725 Tumors

**DOI:** 10.3390/cancers16234052

**Published:** 2024-12-03

**Authors:** Kristina Jansen, Lara Kornfeld, Maximilian Lennartz, Sebastian Dwertmann Rico, Simon Kind, Viktor Reiswich, Florian Viehweger, Ahmed Abdulwahab Bawahab, Christoph Fraune, Natalia Gorbokon, Andreas M. Luebke, Claudia Hube-Magg, Anne Menz, Ria Uhlig, Till Krech, Andrea Hinsch, Frank Jacobsen, Eike Burandt, Guido Sauter, Ronald Simon, Martina Kluth, Stefan Steurer, Andreas H. Marx, Till S. Clauditz, David Dum, Patrick Lebok, Sarah Minner, Christian Bernreuther

**Affiliations:** 1Department of General, Visceral, and Thoracic Surgery, University Medical Center Hamburg-Eppendorf, 20251 Hamburg, Germany; k.jansen@uke.de; 2Institute of Pathology, University Medical Center Hamburg-Eppendorf, 20251 Hamburg, Germany; lara.kornfeld@web.de (L.K.); m.lennartz@uke.de (M.L.); s.dwertmann-rico@uke.de (S.D.R.); s.kind@uke.de (S.K.); v.reiswich@uke.de (V.R.); f.viehweger@uke.de (F.V.); c.fraune@uke.de (C.F.); n.gorbokon@uke.de (N.G.); luebke@uke.de (A.M.L.); c.hube@uke.de (C.H.-M.); r.uhlig@uke.de (R.U.); a.hinsch@uke.de (A.H.); f.jacobsen@uke.de (F.J.); e.burandt@uke.de (E.B.); g.sauter@uke.de (G.S.); m.kluth@uke.de (M.K.); s.steurer@uke.de (S.S.); andreas.marx@klinikum-fuerth.de (A.H.M.); t.clauditz@uke.de (T.S.C.); d.dum@uke.de (D.D.); p.lebok@uke.de (P.L.); s.minner@uke.de (S.M.); c.bernreuther@uke.de (C.B.); 3Department of Basic Medical Sciences, Pathology Division, College of Medicine, University of Jeddah, Jeddah 23218, Saudi Arabia; aabawahab@uj.edu.sa; 4Institute of Pathology, Clinical Center Osnabrueck, 49076 Osnabrueck, Germany; 5Department of Pathology, Academic Hospital Fuerth, 90766 Fuerth, Germany

**Keywords:** CEA, diagnostic marker, tissue microarray, immunohistochemistry, human tumors

## Abstract

Carcinoembryonic antigen is a cell-surface glycoprotein and target for anti-cancer drugs. In this study, more than 15,000 samples from 120 different tumor types were analyzed by immunohistochemistry. CEA expression was found at least occasionally in 65 tumor types, most frequently in colorectal cancers and other gastrointestinal tumors, thyroid gland cancers, and pulmonary adenocarcinomas. Reduced CEA expression was linked to colon cancer aggressiveness. In contrast, aggressiveness cancers of the urinary bladder and breast cancers were characterized by CEA overexpression. We present a comprehensive catalog of tumor types that might benefit from anti-CEA therapies.

## 1. Introduction

Carcinoembryonic antigen (CEA; CEACAM5) is a cell-surface glycoprotein with a role in cell adhesion [1]. CEA is extensively produced in many tissues during fetal development, but even before birth, its expression becomes limited to a limited number of different normal cell types [2]. CEA is overexpressed in various cancers (summarized in [3,4,5,6]). Because CEA is also shed into the blood stream, CEA measurement in the serum is used as a tool for early detection and recurrence monitoring of cancer [7,8]. There are several therapeutic approaches targeting CEA. For example, an ongoing phase I study is investigating the CEA-CD3 bispecific antibody cibisatamab on metastatic CEA-positive colorectal carcinomas. In all, 11% (4/36) of patients treated with cibisatamab monotherapy and 50% (5/10) of patients receiving cibisatamab in combination with a PD-L1-inhibitor showed radiological shrinkage, reflecting its antitumor activity [9,10]. Labetuzumab govitecan, an SN-38 CEA-antibody–drug conjugate, resulted in stable disease in 49% (42/86) of patients with metastatic colorectal cancer in a phase I/II study [11]. A phase I pretargeted radioimmunotherapy trial using anti-CEA-HSG-TF2 showed promising dose optimization for the treatment of CEA-positive metastasized lung carcinomas [12]. Three phase I studies showed that CEA is an auto-antigen that can be safely targeted by CAR T cells, resulting in stable disease in 7/10 patients with metastatic colorectal carcinomas (NCT02349724), 7/14 patients with metastatic gastrointestinal carcinomas (NCT01212887), and 1/6 patients with liver metastases of different adenocarcinomas (NCT01373047) [13]. In addition, vaccination with Ad5 [E1-, E2b-]-CEA(6D)—encoding CEA—induced CEA-specific cell-mediated immune response with antitumor activity in a phase I/II colorectal cancer study [14].

More than 700 studies analyzed the expression of CEA in cancer by immunohistochemistry (IHC) and described CEA positivity in at least 62 different tumor types and subtypes. CEA is considered to occur at a particularly high frequency in colorectal cancers, other gastrointestinal carcinomas, and in pulmonary adenocarcinomas (summarized in [3,4,5,6]). Because CEA expression is rare in hepatocellular carcinoma and in malignant mesothelioma, CEA IHC has been suggested as a tool to distinguish mesothelioma from pulmonary adenocarcinoma and hepatocellular carcinoma from liver metastasis [15,16]. However, the results of previous studies on CEA expression in cancer vary considerably for many tumor types. For example, reported CEA positivity rates ranged from 17% to 100% of pulmonary adenocarcinoma [17,18], 0% to 54% of malignant mesothelioma [15,19], 0% to 100% of hepatocellular carcinoma [20,21], 29% to 100% of colorectal adenocarcinoma [22,23], 25% to 100% of different subtypes of gastric adenocarcinoma [24,25,26,27,28,29], 0% to 100% of cholangiocarcinoma [20,30], 0% to 71% of endometrioid endometrial carcinoma [31,32], 9% to 58% of small-cell carcinoma of the lung [33,34], 0% to 100% of mucinous carcinoma of the ovary [35,36], and 14% to 94% of invasive breast carcinoma of no special type [37,38]. Technical factors, including staining protocols and antibodies used and differences in the definition of thresholds determining positivity, as well as possible selection bias with respect to the analyzed tumors, may have caused these discrepancies. A comprehensive study analyzing as many tumor types as possible under standardized experimental conditions and analysis criteria is, therefore, highly warranted.

Using IHC in a tissue microarray (TMA) format, more than 15,000 tissue samples from 120 different tumor types and subtypes, and 76 non-neoplastic tissues were analyzed, allowing us to better understand the relative importance of CEA expression in various cancers and normal tissues.

## 2. Materials and Methods

### 2.1. Tissue Microarrays (TMAs)

The normal tissue TMA was composed of 8 samples from 8 different donors for each of 76 different normal tissue types (608 samples on one slide). The cancer TMAs contained a total of 15,413 primary tumors from 120 tumor types and subtypes. Detailed histopathological and molecular data on grade, pathological tumor stage (pT), pathological lymph node status (pN), HER2, estrogen receptor (ER),progesterone receptor (PR) status, and mismatch repair protein status were available from subsets of adenocarcinomas of the colon (*n* = 2351), the pancreas (*n* = 598), and the stomach (*n* = 327); invasive breast carcinomas of no special type (*n* = 1208); urothelial bladder carcinomas (*n* = 1663); endometrioid endometrial carcinomas (*n* = 182); and endometrioid (*n* = 40) and serous carcinoma of the ovary (*n* = 369). Clinical follow-up data (overall survival) were available from 877 breast cancer patients, with a median follow-up time of 49 months (range 1–88 months). The composition of both normal and cancer TMAs is described in detail in the Section 3. All samples were from the archives of the Institutes of Pathology, University Hospital of Hamburg, Germany; the Institute of Pathology, Clinical Center Osnabrueck, Germany; or the Department of Pathology, Academic Hospital, Fuerth, Germany. Tissues were fixed in 4% buffered formalin and then embedded in paraffin. The TMA manufacturing process was described earlier in detail [39,40]. In brief, one tissue spot (diameter: 0.6 mm) was transmitted from a cancer-containing donor block in an empty recipient paraffin block. The use of archived remnants of diagnostic tissues for manufacturing of TMAs, and their analysis for research purposes, as well as patient data analyses, have been approved by local laws (HmbKHG, §12) and by the local ethics committee (Ethics commission Hamburg, WF-049/09). All work has been carried out in compliance with the Helsinki Declaration.

### 2.2. Immunohistochemistry (IHC)

Freshly prepared TMA sections were immunostained on one day in one experiment. Slides were deparaffinized with xylol, rehydrated through a graded alcohol series and exposed to heat-induced antigen retrieval for 5 min in an autoclave at 121 °C in pH 9.0 DakoTarget Retrieval Solution^TM^ (Agilent, Santa Clara, CA, USA; #S2367). Endogenous peroxidase activity was blocked with Dako Peroxidase Blocking Solution^TM^ (Agilent #52023) for 10 min. Primary antibody specific against CEA protein (rabbit recombinant, MSVA-465R, MS Validated Antibodies, Hamburg, Germany; #2563-465R) was applied at 37 °C for 60 min at a dilution of 1:150. Bound antibody was visualized using the EnVision Kit^TM^ (Agilent #K5007) according to the manufacturer’s directions. For the purpose of antibody validation, the normal tissue TMA was also analyzed by an additional CEA antibody (monoclonal mouse, II-7, Agilent #M7072) on a Agilent DAKO autostainer Link48 according to a protocol suggested by Agilent DAKO. The sections were counterstained with hemalaun. For tumor tissues, the percentage of CEA-positive tumor cells was estimated and the staining intensity was semi-quantitatively recorded (0, 1+, 2+, 3+). For statistical analyses, the staining results were categorized into four groups as follows: negative, no staining at all; weak staining, staining intensity of 1+ in ≤70% or staining intensity of 2+ in ≤30% of tumor cells; moderate staining, staining intensity of 1+ in >70%, staining intensity of 2+ in >30% but in ≤70%, or staining intensity of 3+ in ≤30% of tumor cells; and strong staining: staining intensity of 2+ in >70% or staining intensity of 3+ in >30% of tumor cells.

### 2.3. Statistics

Statistical calculations were performed with JMP 16 software (SAS Institute Inc., Cary, NC, USA). Contingency tables and the chi^2^ test were performed to search for associations between CEA immunostaining and tumor phenotype. Overall survival curves were calculated according to Kaplan–Meier. The Log-Rank test was applied to detect significant differences between groups. A *p*-value of ≤0.05 was considered to be statistically significant.

## 3. Results

### 3.1. Technical Issues

In our TMA analysis, a total of 13,725 (89.0%) of 15,413 tumor samples and over three samples for each normal tissue category were analyzable. Non-analyzable samples showed absence of tissue or lack of indisputable tumor cells in their respective TMA spots.

### 3.2. CEA in Normal Tissues

For MSVA-465R, a moderate-to-strong CEA immunostaining was seen, particularly in the upper layers of the non-keratinizing (but not keratinizing) squamous epithelium, irrespective of the location. In the skin, eccrine glands showed a luminal membrane staining. Strong CEA staining occurred in the transitional epithelium of the anal canal and in most cells of Hassall’s corpuscles in the thymus. The strongest CEA staining in the gastrointestinal tract was seen in the colorectal mucosa, where the staining was strongest in the surface cells and the upper half of crypts. Stomach mucosa surface cell layers showed a moderate-to-strong CEA positivity, while glands were either negative or much less stained. Duodenum and small intestine showed a moderate staining of a subset of goblet cells, primarily at the surface epithelium, while deeper glands were mostly negative. A weak-to-moderate staining occurred at the apical membranes of a fraction of (mainly) mucinous cells in salivary glands. A variable CEA staining occurred in respiratory epithelium, primarily in goblet cells. A significant fraction of pneumocytes also showed an apical membrane positivity. Urothelium was usually CEA-negative but occasionally showed a variable weak staining of umbrella cells. Representative images are shown in Figure 1. All of these cell types were also found to be CEA-positive by using the monoclonal mouse antibody IL-7 (Appendix A). CEA staining was absent in mesenchymal, lymphatic, and hematopoietic cells; skin; liver; adrenal gland; kidney; gall bladder epithelium; Brunner glands; prostate; seminal vesicle; epididymis; testis; breast; placenta; endocervix; endometrium; ovary with corpus luteum and follicular cysts; thyroid; hypophysis; cerebrum; and in the cerebellum (Appendix A).

### 3.3. CEA in Cancer

All cancers were analyzed with MSVA-465R. CEA immunostaining was detectable in 4323 (31.5%) of the 13,725 analyzable tumors. Weak CEA immunostaining was seen in 1076 tumors (7.8%), moderate in 425 (3.1%), and strong in 2822 (20.6%). Of the 120 tumor categories included, CEA positivity was found in 65 (54.2%) while 49 (40.8%) tumor categories showed at least one case of strong positivity (Table 1).

Examples for particularly high positivity rates and high levels of expression were colorectal adenomas (100% positive) and adenocarcinomas (98.7%), other gastrointestinal adenocarcinomas (61.1–80.3%), medullary thyroid carcinomas (96.3%), adenocarcinoma of the lung (73.7%), mucinous carcinomas of the ovary (79.8%) and the breast (43.2%), squamous cell carcinomas of different sites of origin (30.2–69.1%) as well as small-cell carcinomas of the lung (64.3%), the prostate (50.0%), and the urinary bladder (38.9%). In many of these tumor types, CEA expression was often stronger than in the corresponding normal tissues. Figure 2 shows representative images. Figure 3 gives a graphical representation of the ranking order of CEA-positive and strongly positive tumors. Associations between CEA immunostaining and histopathological features are shown in Table 2. High CEA expression was associated with advanced pT stage (*p* < 0.0001) and muscle-invasive growth (*p* < 0.0001) in urinary bladder cancer. In invasive breast carcinomas of no special type, high CEA expression was linked to ER positivity (*p* = 0.0005) and HER2 positivity (*p* < 0.0001) but was unrelated to grade, pT, pN, and overall survival (*p* = 0.2520, Figure 4). Reduced CEA expression was associated with defective mismatch repair status (dMMR; *p* < 0.0001), BRAF V600E mutations (*p* = 0.0498), and tumor localization in the right colon (*p* = 0.0024) in colorectal adenocarcinoma. There was no association of the CEA expression level to histopathological, molecular, or clinical tumor characteristics in pancreatic and gastric adenocarcinomas, or endometroid endometrium carcinoma. A combined analysis of 524 squamous cell carcinomas from nine different sites showed a link between HPV infection and CEA positivity (*p* = 0.0281; Appendix A) but subgroup analysis by organs of origins did not show significant associations between HPV status and CEA expression levels.

## 4. Discussion

The importance of CEA as a cancer-related protein is reflected by more than 3000 Medline articles meeting the search terms “CEA + immunohistochemistry + cancer” (pubmed_January 2023). The articles generally agree on colorectal adenocarcinoma being the cancer entity most commonly CEA-positive, but data on the prevalence of CEA expression vary considerably in other tumor types. Based on our highly standardized analysis of 120 tumor types and subtypes, we were able to create a ranking order of human tumors according to the prevalence of CEA expression (Figure 3), which allows for a direct comparison of CEA positivity obtained under identical experimental conditions. A collection of data from the literature (Figure 5, Appendix A) demonstrates that such information could not easily be retrieved from the literature.

Although—in line with our data—multiple studies showed high rates of CEA positivity in tumor types arising from epithelial tissues, including gastrointestinal adenocarcinomas, medullary carcinomas of the thyroid, adenocarcinomas of the lung, and squamous cell carcinomas of various origins, as well as in small-cell carcinomas, there were also numerous studies suggesting much lower positivity rates in these tumor types. Moreover, our data identified low CEA positivity rates in several tumor entities for which high CEA expression levels were suggested in multiple earlier studies, such as, for example, in high-grade serous carcinomas of the ovary [41], or prostatic adenocarcinomas [42]. That high levels of CEA expression can be observed in many different tumor entities is consistent with data summarized in The Cancer Genome Atlas (TCGA) database (www.cancer.gov, data available from https://www.proteinatlas.org/ENSG00000105388-CEACAM5/summary/rna, accessed on 21 November 2024).

Considering the high expression levels of CEA in various tumor entities and the relatively low CEA levels in normal tissues, CEA represents an attractive therapeutic target. However, previous attempts at targeting CEA by using humanized anti-CEA antibodies showed disappointing results. It is assumed that in cancers with high serum CEA levels, the therapeutic antibodies are prevented from reaching the tumor cells, as they bind to circulating CEA (summarized in [43]). Promising recent approaches include developing a vaccination (summarized in [44]) and the generation of CAR-T cells (summarized in [45]) targeting CEA. CEA-targeting drugs are currently studied in more than 200 phase I and II clinical trials. Included in these studies are colorectal, breast, esophageal, stomach, lung, gastric, and pancreatic carcinomas as well as CEA-positive tumors regardless of their sites of origin (www.clinicaltrials.gov). If one of these treatment approaches should become clinically available, our ranking order of tumors based on their CEA positivity rates could help to determine the tumor entities for which such approaches would be most beneficial.

Our data support two important diagnostic applications of CEA IHC. The complete CEA-negativity of all our 90 malignant mesotheliomas while 74% of the 179 pulmonary adenocarcinomas were CEA-positive supports the use of CEA IHC for the distinction of malignant mesothelioma from pulmonary adenocarcinoma. Panels that have been proposed for this application also include calretinin, D2-40, WT1, cytokeratin 5/6, D2-40, EpCAM, TTF-1, Ber-EP4/MOC31, and Napsin A [46]. Total absence of CEA staining in all 50 hepatocellular carcinomas analyzed in this study supports the concept of using CEA as a marker for the distinction of primary tumors from metastases in the liver [16]. This is all the more useful, as carcinomas from the entire gastrointestinal tract—the most common source of liver metastases—were often CEA-positive. The high rate of CEA-positive hepatocellular carcinomas described in several previous studies is likely due to the use of less specific and/or polyclonal CEA antibodies, which leads to positivity rates between 15–100% [30,47,48,49]. Previous studies with monoclonal CEA antibodies identified a CEA positivity in 0–55% of hepatocellular carcinomas [35,48,49,50].

For several of our tumor categories, the large number of tumors analyzed allowed an analysis of the potential clinical significance of CEA expression. The strong link between a reduced CEA expression and dMMR in colorectal cancer, in which 96% of mismatch repair-proficient (pMMR) but only 69% of dMMR cancers showed strong CEA staining is in agreement with data from Schiemann et al. [51] finding lower CEA serum levels in colorectal cancer patients with hereditary non-polyposis colorectal carcinomas than found in those with sporadic carcinomas. However, another study showed only a marginal difference between dMMR (67% negative) and pMMR (58%) colorectal carcinomas [52] while three further studies found no difference in CEA immunostaining between dMMR and pMMR carcinomas [53,54,55]. Given the abundance of DNA mutations in microsatellite instable/dMMR cancers, it appears possible that functionally relevant DNA alterations of genes that directly or indirectly regulate CEA expression may cause the reduced CEA expression in a subset of dMMR cancers. The absence of clear-cut associations with histopathological parameters of tumor aggressiveness in several tumor entities and the lack of a prognostic significance in invasive breast cancers of no special type strongly argues against a strong and clinically relevant prognostic role of CEA expression in cancer and suggests that CEA upregulation may parallel tumor development in various cancer types. This interpretation is supported by controversial results of previous studies. Although there were more than 18 studies suggesting an unfavorable tumor phenotype or poor prognosis for tumors with high CEA expression, there were at least 24 studies denying such a role of CEA expression (Appendix A). Similarly controversial data were seen for specific tumor entities. For example, in colorectal adenocarcinomas, eight studies linked high CEA expression to unfavorable tumor features and/or prognosis while six studies found no association between CEA immunostaining and tumor phenotype and/or prognosis (Appendix A).

CEA mainly functions as a serum marker for colorectal cancer. Serial CEA serum measurements were recommended in a 2014 update of the European group on tumor marker guidelines for postoperative surveillance of UICC stage II and III patients considered for surgical resection or systemic therapy in case of distant metastasis and for monitoring response to treatment in advanced disease [56]. High preoperative [57] and postoperative [58] serum CEA levels have also been suggested as a prognostic tool in colorectal cancer. Given the abundant and homogeneous expression of CEA in most colorectal cancers, this is likely due to a link between CEA serum levels, tumor burden, and residual disease. Our findings demonstrate that similarly high levels of tumoral CEA expression can occur in many other tumor types, at least in a significant fraction of cases. Given the close to 100% prevalence of CEA expression in colorectal cancer, serological CEA monitoring obviously does not require CEA analysis in tumor tissue. It appears possible, however, that an immunohistochemical tumor tissue analysis documenting high-level CEA expression in a non-colorectal cancer could identify individual patients for whom CEA serum levels would also be suited for monitoring response to therapy and early detection of tumor relapse. Various studies suggested a clinical utility of CEA serum assessment in gastric [59], pancreatic [60], non-small-cell lung [61], and breast cancer [62], although the majority of these studies did not improve patient selection by immunohistochemical CEA analysis of the tumor tissue.

Regarding the large scale of our study, a particular emphasis was placed on the validation of our staining procedure, which was conducted according to the methods proposed by the International Working Group for Antibody Validation (IWGAV). In these guidelines, a comparison with expression data obtained by an additional independent method or a second independent antibody are suggested [63]. Both methods were applied in this study. CEA IHC in 76 different normal tissues was first compared with RNA expression data compiled from the Human Protein Atlas RNA-seq tissue dataset [64], the Functional Annotation of the Mammalian Genome (FANTOM5) project [65,66], and the Genotype-Tissue Expression (GTEx) project [67]. Our normal tissue analysis revealed CEA immunostaining in 10 of 12 organs for which RNA expression had been described (rectum, colon, appendix, small intestine, duodenum, stomach, salivary gland, tonsil, cervix, and lung). Only two organs with documented RNA expression did not show CEA immunostaining (esophagus, smooth muscle). As esophagus and smooth muscle can both be adjacent to CEA-expressing epithelium, these RNA findings may reflect contaminations. In the—CEA RNA negative—thymus and the skin, CEA immunostaining was limited to very small subpopulations that might have been missed in RNA analyses of entire organs. For two other CEA IHC positive tissues (bronchus, anal canal), RNA data were unavailable. The comparison with an independent second antibody (IL-7) confirmed CEA protein expression in bronchus, anal canal, thymus, and the skin, as well as in all other cell types identified as CEA-positive by MSVA-465R.

## 5. Conclusions

The comprehensive list of CEA positivity across human tumor types shows that CEA is abundantly expressed in a broad range of epithelial neoplasms and serves as a basis for potential future research. Our data identify these tumor entities where most CEA-positive cancers might benefit from CEA serum monitoring and anti-CEA therapies. However, the level of CEA expression does not appear to be largely related to cancer aggressiveness.

## Figures and Tables

**Figure 1 cancers-16-04052-f001:**
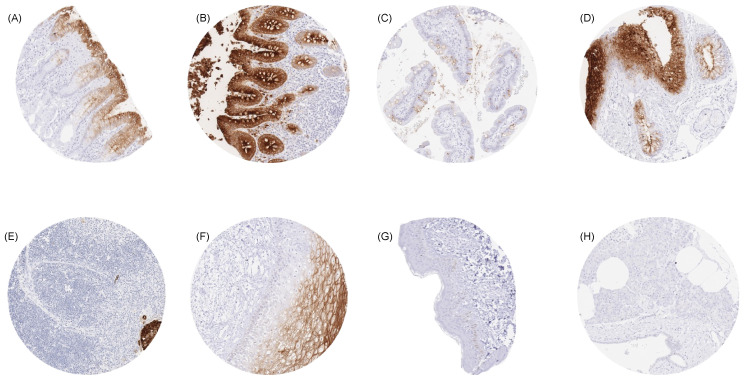
CEA immunostaining of normal tissues. A membranous and cytoplasmic CEA staining of variable intensity is seen in surface epithelial cells of the stomach (**A**), epithelial cells (predominantly at the surface) of the colon (**B**), goblet cells of the small intestine (**C**), respiratory epithelial cells (**D**), Hassal’s corpuscles of the thymus (**E**), and in superficial cell layers of the squamous epithelium of the cervix uteri (**F**). CEA staining is absent in tissues from the epidermis of the skin (**G**) and in the pancreas (**H**). Original magnifications 10×, spot size 600 μm.

**Figure 2 cancers-16-04052-f002:**
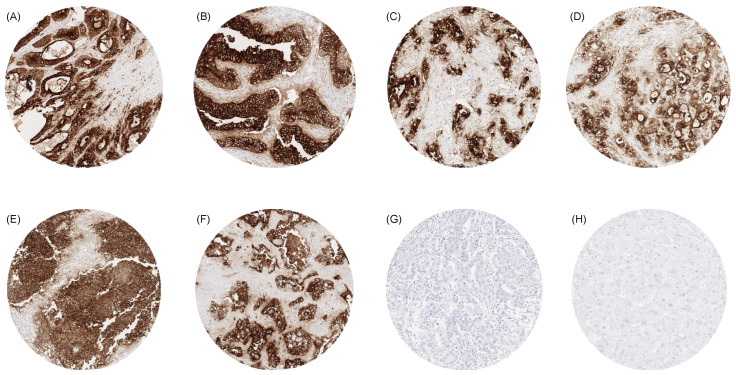
CEA immunostaining in cancer. The panels show a strong CEA staining in an adenocarcinoma of the colon (**A**), an adenocarcinoma of the esophagus (**B**), a ductal adenocarcinoma of the pancreas (**C**), a cholangiocellular carcinoma of the liver (**D**), a small-cell neuroendocrine carcinoma of the lung (**E**), and an adenocarcinoma of the lung (**F**). CEA staining is lacking in a malignant mesothelioma of the pleura (**H**) and a hepatocellular carcinoma in the liver (**G**). Original magnifications 10×, spot size 600 μm.

**Figure 3 cancers-16-04052-f003:**
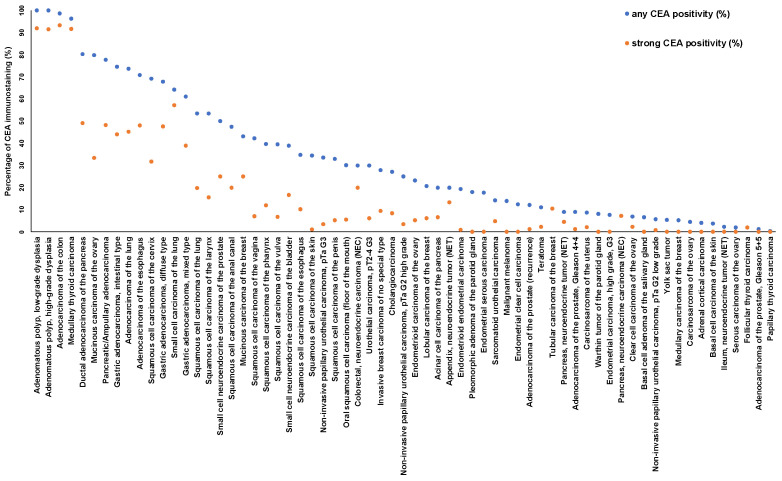
Ranking order of CEA immunostaining in cancers. Both the percentage of positive cases (blue dots) and the percentage of strongly positive cases (orange dots) are shown.

**Figure 4 cancers-16-04052-f004:**
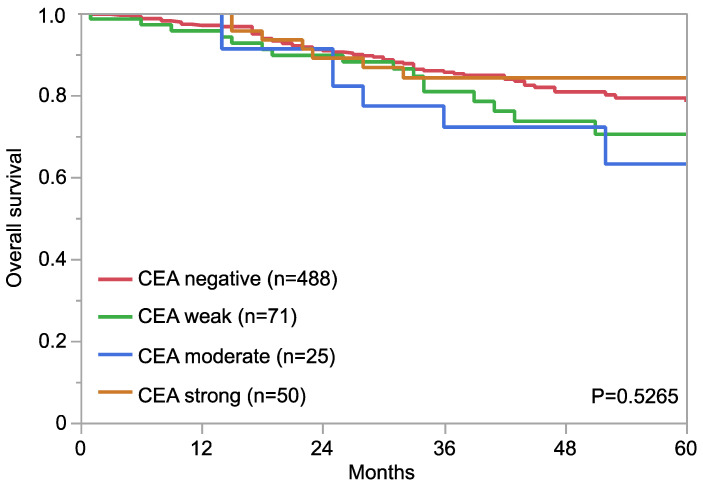
CEA immunostaining and patient prognosis in invasive breast carcinoma of no special type.

**Figure 5 cancers-16-04052-f005:**
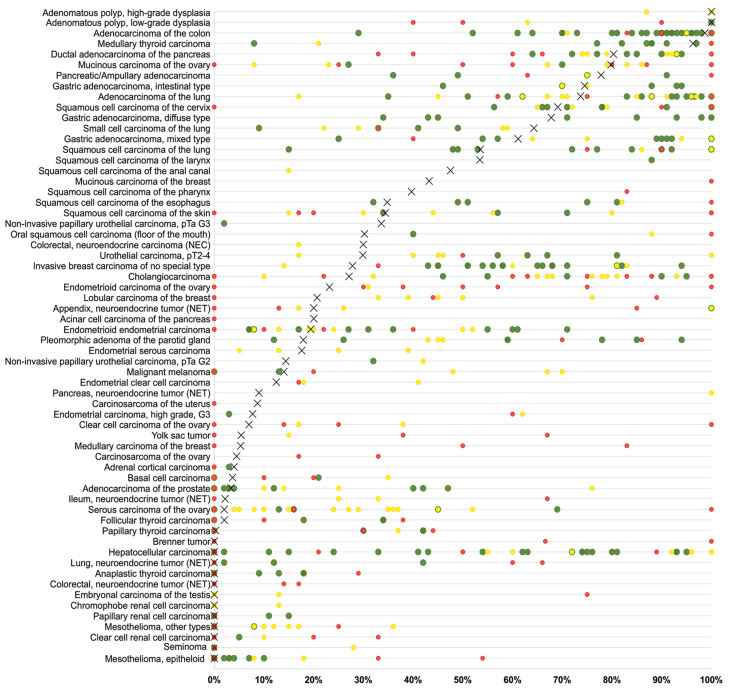
Comparison of CEA expression with previous works in the literature. An “X” represents the proportion of CEA-positive cancers in the present study, dots indicate the frequencies reported in the literature for comparison: studies with ≤10 tumors analyzed are marked with red dots, studies with 11 to 25 tumors analyzed are marked with yellow dots, and green dots mark studies with >25 tumors analyzed.

**Table 1 cancers-16-04052-t001:** CEA immunostaining in human tumors. Abbreviations: int. = number of interpretable samples, neg. = negative, mod. = moderate, str. = strong.

			CEA Immunostaining Result				
	Tumor Entity	On TMA (*n*)	Int. (*n*)	Neg. (%)	Weak (%)	Mod. (%)	Str. (%)
Tumors of the skin	Pilomatricoma	35	23	100.0	0.0	0.0	0.0
	Basal cell carcinoma of the skin	88	80	96.3	3.8	0.0	0.0
	Benign nevus	29	29	100.0	0.0	0.0	0.0
	Squamous cell carcinoma of the skin	90	90	65.6	31.1	2.2	1.1
	Malignant melanoma	46	43	86.0	9.3	4.7	0.0
	Merkel cell carcinoma	46	42	100.0	0.0	0.0	0.0
Tumors of the head and neck	Squamous cell carcinoma of the larynx	109	103	46.6	30.1	7.8	15.5
	Squamous cell carcinoma of the pharynx	60	58	60.3	22.4	5.2	12.1
	Oral squamous cell carcinoma (floor of the mouth)	130	126	69.8	17.5	7.1	5.6
	Pleomorphic adenoma of the parotid gland	50	39	82.1	15.4	2.6	0.0
	Warthin tumor of the parotid gland	49	49	91.8	8.2	0.0	0.0
	Basal cell adenoma of the salivary gland	15	15	93.3	6.7	0.0	0.0
Tumors of the lung, pleura, and thymus	Adenocarcinoma of the lung	196	179	26.3	19.6	8.9	45.3
	Squamous cell carcinoma of the lung	80	71	46.5	28.2	5.6	19.7
	Small-cell carcinoma of the lung	16	14	35.7	0.0	7.1	57.1
	Mesothelioma, epithelioid	39	28	100.0	0.0	0.0	0.0
	Mesothelioma, biphasic	76	62	100.0	0.0	0.0	0.0
	Thymoma	29	28	100.0	0.0	0.0	0.0
	Lung, neuroendocrine tumor (NET)	19	18	100.0	0.0	0.0	0.0
Tumors of the female genital tract	Squamous cell carcinoma of the vagina	78	71	57.7	25.4	9.9	7.0
	Squamous cell carcinoma of the vulva	130	119	60.5	30.3	2.5	6.7
	Squamous cell carcinoma of the cervix	129	123	30.9	23.6	13.8	31.7
	Endometrioid endometrial carcinoma	236	227	80.6	18.1	0.4	0.9
	Endometrial serous carcinoma	82	68	82.4	17.6	0.0	0.0
	Carcinosarcoma of the uterus	48	46	91.3	4.3	2.2	2.2
	Endometrial carcinoma, high grade, G3	13	13	92.3	7.7	0.0	0.0
	Endometrial clear cell carcinoma	8	8	87.5	12.5	0.0	0.0
	Endometrioid carcinoma of the ovary	110	95	76.8	15.8	2.1	5.3
	Serous carcinoma of the ovary	559	499	98.0	1.8	0.2	0.0
	Mucinous carcinoma of the ovary	96	84	20.2	28.6	17.9	33.3
	Clear cell carcinoma of the ovary	50	43	93.0	4.7	0.0	2.3
	Carcinosarcoma of the ovary	47	44	95.5	4.5	0.0	0.0
	Brenner tumor	9	9	100.0	0.0	0.0	0.0
Tumors of the breast	Invasive breast carcinoma of no special type	1345	1042	72.2	13.1	5.4	9.4
	Lobular carcinoma of the breast	293	193	79.3	8.8	5.7	6.2
	Medullary carcinoma of the breast	26	19	94.7	5.3	0.0	0.0
	Tubular carcinoma of the breast	27	19	89.5	0.0	0.0	10.5
	Mucinous carcinoma of the breast	58	44	56.8	11.4	6.8	25.0
	Phyllodes tumor of the breast	50	43	100.0	0.0	0.0	0.0
Tumors of the digestive system	Adenomatous polyp, low-grade dysplasia	50	50	0.0	0.0	8.0	92.0
	Adenomatous polyp, high-grade dysplasia	50	47	0.0	0.0	8.5	91.5
	Adenocarcinoma of the colon	1882	1773	1.3	2.1	3.3	93.3
	Gastric adenocarcinoma, diffuse type	176	149	32.2	12.1	8.1	47.7
	Gastric adenocarcinoma, intestinal type	174	161	25.5	15.5	14.9	44.1
	Gastric adenocarcinoma, mixed type	62	54	38.9	13.0	9.3	38.9
	Adenocarcinoma of the esophagus	83	79	29.1	16.5	6.3	48.1
	Squamous cell carcinoma of the esophagus	76	69	65.2	14.5	10.1	10.1
	Squamous cell carcinoma of the anal canal	89	80	52.5	23.8	3.8	20.0
	Cholangiocarcinoma	113	107	72.9	12.1	6.5	8.4
	Hepatocellular carcinoma	50	50	100.0	0.0	0.0	0.0
	Ductal adenocarcinoma of the pancreas	612	553	19.7	20.3	10.8	49.2
	Pancreatic/Ampullary adenocarcinoma	89	81	22.2	18.5	11.1	48.1
	Acinar cell carcinoma of the pancreas	16	15	80.0	6.7	6.7	6.7
	Gastrointestinal stromal tumor (GIST)	50	46	100.0	0.0	0.0	0.0
	Appendix, neuroendocrine tumor (NET)	22	15	80.0	6.7	0.0	13.3
	Colorectal, neuroendocrine tumor (NET)	12	12	100.0	0.0	0.0	0.0
	Ileum, neuroendocrine tumor (NET)	49	45	97.8	2.2	0.0	0.0
	Pancreas, neuroendocrine tumor (NET)	97	89	91.0	2.2	2.2	4.5
	Colorectal, neuroendocrine carcinoma (NEC)	12	10	70.0	10.0	0.0	20.0
	Gallbladder, neuroendocrine carcinoma (NEC)	4	4	100.0	0.0	0.0	0.0
	Pancreas, neuroendocrine carcinoma (NEC)	14	14	92.9	0.0	0.0	7.1
Tumors of the urinary system	Non-invasive papillary urothelial ca., pTa G2 low grade	177	141	94.3	5.0	0.0	0.7
	Non-invasive papillary urothelial ca., pTa G2 high grade	141	116	75.0	20.7	0.9	3.4
	Non-invasive papillary urothelial carcinoma, pTa G3	187	116	66.4	26.7	3.4	3.4
	Urothelial carcinoma, pT2-4 G3	1206	835	70.1	19.2	4.7	6.1
	Small-cell neuroendocrine carcinoma of the bladder	20	18	61.1	16.7	5.6	16.7
	Sarcomatoid urothelial carcinoma	25	21	85.7	9.5	0.0	4.8
	Clear cell renal cell carcinoma	857	835	100.0	0.0	0.0	0.0
	Papillary renal cell carcinoma	255	242	100.0	0.0	0.0	0.0
	Clear cell (tubulo) papillary renal cell carcinoma	21	21	100.0	0.0	0.0	0.0
	Chromophobe renal cell carcinoma	131	127	100.0	0.0	0.0	0.0
	Oncocytoma of the kidney	177	165	100.0	0.0	0.0	0.0
Tumors of the male genital organs	Adenocarcinoma of the prostate, Gleason 3 + 3	83	83	100.0	0.0	0.0	0.0
	Adenocarcinoma of the prostate, Gleason 4 + 4	80	78	91.0	7.7	0.0	1.3
	Adenocarcinoma of the prostate, Gleason 5 + 5	85	85	98.8	1.2	0.0	0.0
	Adenocarcinoma of the prostate (recurrence)	258	248	87.9	10.1	0.8	1.2
	Small-cell neuroendocrine carcinoma of the prostate	19	16	50.0	18.8	6.3	25.0
	Seminoma	621	613	100.0	0.0	0.0	0.0
	Embryonal carcinoma of the testis	50	44	100.0	0.0	0.0	0.0
	Yolk sac tumor	50	37	94.6	5.4	0.0	0.0
	Teratoma	50	45	88.9	4.4	4.4	2.2
	Squamous cell carcinoma of the penis	80	76	67.1	19.7	7.9	5.3
Tumors of endocrine organs	Adenoma of the thyroid gland	113	107	100.0	0.0	0.0	0.0
	Papillary thyroid carcinoma	391	384	99.7	0.3	0.0	0.0
	Follicular thyroid carcinoma	154	152	98.0	0.0	0.0	2.0
	Medullary thyroid carcinoma	111	107	3.7	0.9	3.7	91.6
	Anaplastic thyroid carcinoma	45	42	100.0	0.0	0.0	0.0
	Adrenal cortical adenoma	50	45	100.0	0.0	0.0	0.0
	Adrenal cortical carcinoma	26	25	96.0	0.0	4.0	0.0
	Pheochromocytoma	50	49	100.0	0.0	0.0	0.0
Tumors of hematopoetic and lymphoid tissues	Hodgkin lymphoma	103	98	100.0	0.0	0.0	0.0
	Small lymphocytic lymphoma, B-cell type (B-SLL/B-CLL)	50	50	100.0	0.0	0.0	0.0
	Diffuse large B cell lymphoma (DLBCL)	113	113	100.0	0.0	0.0	0.0
	Follicular lymphoma	88	88	100.0	0.0	0.0	0.0
	T-cell non-Hodgkin lymphoma	25	25	100.0	0.0	0.0	0.0
	Mantle cell lymphoma	18	18	100.0	0.0	0.0	0.0
	Marginal zone lymphoma	16	16	100.0	0.0	0.0	0.0
	Diffuse large B-cell lymphoma (DLBCL) in the testis	16	16	100.0	0.0	0.0	0.0
	Burkitt lymphoma	5	3	100.0	0.0	0.0	0.0
Tumors of soft tissue and bone	Tenosynovial giant cell tumor	45	44	100.0	0.0	0.0	0.0
	Granular cell tumor	53	45	100.0	0.0	0.0	0.0
	Leiomyoma	50	49	100.0	0.0	0.0	0.0
	Leiomyosarcoma	87	87	100.0	0.0	0.0	0.0
	Liposarcoma	132	123	100.0	0.0	0.0	0.0
	Malignant peripheral nerve sheath tumor (MPNST)	13	13	100.0	0.0	0.0	0.0
	Myofibrosarcoma	26	26	100.0	0.0	0.0	0.0
	Angiosarcoma	73	68	100.0	0.0	0.0	0.0
	Angiomyolipoma	91	89	100.0	0.0	0.0	0.0
	Dermatofibrosarcoma protuberans	21	18	100.0	0.0	0.0	0.0
	Ganglioneuroma	14	14	100.0	0.0	0.0	0.0
	Kaposi sarcoma	8	5	100.0	0.0	0.0	0.0
	Neurofibroma	117	114	100.0	0.0	0.0	0.0
	Sarcoma, not otherwise specified (NOS)	74	71	100.0	0.0	0.0	0.0
	Paraganglioma	41	41	100.0	0.0	0.0	0.0
	Ewing sarcoma	23	16	100.0	0.0	0.0	0.0
	Rhabdomyosarcoma	6	6	100.0	0.0	0.0	0.0
	Schwannoma	121	118	100.0	0.0	0.0	0.0
	Synovial sarcoma	12	11	100.0	0.0	0.0	0.0
	Osteosarcoma	43	37	100.0	0.0	0.0	0.0
	Chondrosarcoma	38	21	100.0	0.0	0.0	0.0

**Table 2 cancers-16-04052-t002:** CEA immunostaining and cancer phenotype.

			CEA Immunostaining Result				
		*n*	Negative (%)	Weak (%)	Moderate (%)	Strong (%)	*p*
Invasive breast carcinoma of no special type	pT1	491	75.2	12.0	5.3	7.5	0.5044
	pT2	369	70.5	13.6	5.1	10.8	
	pT3–4	80	68.8	15.0	3.8	12.5	
	G1	153	78.4	10.5	5.2	5.9	0.1972
	G2	472	69.9	14.0	6.6	9.5	
	G3	345	74.2	12.5	3.5	9.9	
	pN0	269	71.4	18.2	5.6	4.8	0.6430
	pN+	171	66.7	20.5	8.2	4.7	
	pM0	137	70.1	16.8	5.1	8.0	0.8864
	pM1	93	66.7	17.2	7.5	8.6	
	HER2 negative	771	74.2	11.9	5.1	8.8	0.0158
	HER2 positive	113	62.8	23.0	3.5	10.6	
	ER negative	184	83.7	8.7	2.2	5.4	0.0005
	ER positive	662	68.9	15.4	6.0	9.7	
	PR negative	351	74.6	12.8	4.0	8.5	0.6741
	PR positive	527	71.5	13.7	5.5	9.3	
	Non-triple negative	701	68.9	15.4	5.6	10.1	<0.0001
	Triple negative	123	91.1	4.1	2.4	2.4	
Urothelial bladder carcinoma	pTa G2 low	143	93.0	5.6	0.0	1.4	<0.0001
	pTa G2 high	116	75.0	20.7	0.9	3.4	
	pTa G3	116	66.4	26.7	3.4	3.4	
	pT2	136	69.1	20.6	2.9	7.4	0.0167
	pT3	216	69.0	16.2	10.6	4.2	
	pT4	97	76.3	17.5	2.1	4.1	
	G2	22	68.2	22.7	4.5	4.5	0.9255 *
	G3	427	70.7	17.6	6.6	5.2	
	pN0	253	72.7	17.0	5.5	4.7	0.4985 *
	pN+	170	66.5	20.6	8.2	4.7	
Adenocarcinoma of the pancreas	pT1	14	21.4	14.3	7.1	57.1	0.4277
	pT2	70	10.0	27.1	11.4	51.4	
	pT3	373	20.6	19.0	10.7	49.6	
	pT4	29	17.2	27.6	17.2	37.9	
	G1	16	18.8	18.8	12.5	50.0	0.9969
	G2	344	18.9	19.5	11.3	50.3	
	G3	104	19.2	22.1	9.6	49.0	
	pN0	106	19.8	17.9	13.2	49.1	0.7820
	pN+	379	18.5	21.4	10.6	49.6	
	R0	249	18.5	20.9	11.6	49.0	0.8254
	R1	198	18.2	20.7	9.1	52.0	
	MMR proficient	440	19.3	20.7	11.8	48.2	0.2974
	MMR deficient	4	0.0	50.0	25.0	25.0	
Adenocarcinoma of the stomach	pT1–2	57	31.6	8.8	10.5	49.1	0.6383
	pT3	110	33.6	11.8	9.1	45.5	
	pT4	106	24.5	16.0	13.2	46.2	
	pN0	68	27.9	8.8	19.1	44.1	0.1081
	pN+	204	30.9	13.7	8.3	47.1	
	MMR proficient	232	30.6	13.8	9.9	45.7	0.0899
	MMR deficient	37	18.9	16.2	24.3	40.5	
Endometrioid endometrial carcinoma	pT1	115	79.1	20.0	0.0	0.9	0.3498
	pT2	24	83.3	12.5	4.2	0.0	
	pT3–4	35	88.6	11.4	0.0	0.0	
	pN0	50	80.0	20.0	0.0	0.0	0.7254
	pN+	30	76.7	23.3	0.0	0.0	
Endometrioid carcinoma of the ovary	pT1	23	82.6	17.4	0.0	0.0	0.2037
	pT2	5	60.0	20.0	0.0	20.0	
	pT3	6	50.0	33.3	0.0	16.7	
	pN0	21	71.4	28.6	0.0	0.0	0.0516
	pN1	8	62.5	12.5	0.0	25.0	
Serous carcinoma of the ovary	pT1	32	100.0	0.0	0.0	0.0	0.2000
	pT2	44	95.5	4.5	0.0	0.0	
	pT3	255	99.2	0.8	0.0	0.0	
	pN0	83	100.0	0.0	0.0	0.0	0.1147
	pN1	163	98.2	1.8	0.0	0.0	
Adenocarcinoma of the colon	pT1	66	3.0	0.0	3.0	93.9	0.5369
	pT2	336	1.2	2.4	2.7	93.8	
	pT3	936	1.3	1.7	3.4	93.6	
	pT4	345	1.2	3.5	2.9	92.5	
	pN0	877	1.0	1.5	3.2	94.3	0.1474
	pN+	790	1.6	2.9	3.0	92.4	
	V0	1225	1.1	2.2	3.1	93.6	0.6785
	V1	432	1.9	2.1	3.2	92.8	
	L0	629	1.1	1.3	2.7	94.9	0.1296
	L1	1013	1.5	2.8	3.5	92.3	
	Right side	454	2.4	3.3	4.6	89.6	0.0024
	Left side	1237	0.8	1.8	2.6	94.8	
	MMR proficient	1162	0.6	1.3	2.0	96.1	<0.0001
	MMR deficient	88	5.7	11.4	13.6	69.3	
	RAS wildtype	468	0.9	2.8	3.0	93.4	0.7409
	RAS mutation	355	1.1	2.3	2.0	94.6	
	BRAF wildtype	124	0.8	3.2	1.6	94.4	0.0498
	BRAF V600E mutation	21	0.0	9.5	14.3	76.2	

* only in pT2–4 urothelial bladder carcinomas. Abbreviations: pT, pathological tumor stage; G, Grade; pN, pathological lymph node status; pM, pathological status of distant metastasis; R, resection margin status; V, venous invasion; L, lymphatic invasion; PR, progesterone receptor; ER, estrogen receptor; MMR, mismatch repair.

## Data Availability

All data generated or analyzed during this study are included in this published article.

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
