# Peer review of "Carcinoembryonic Antigen Expression in Human Tumors: A Tissue Microarray Study on 13,725 Tumors"

_cancers, 2024, doi:10.3390/cancers16234052_

Round 1
Reviewer 1 Report
Comments and Suggestions for Authors
Dear authors, greeting for your manuscript.
It well summarizes and describes in a very detailed way CEA expression in cancer. It confirms and encurages to use CEA as serum marker for diagnostic and monitoring in cancer disease.
Figures and tables also in manuscript both in supplementary data are exaustive and of high quality.
References are representative of international scientific literature and updated.
I accept this manuscript for publish in this way.
Thank you
Author Response
Reviewer 1
Dear authors, greeting for your manuscript.
It well summarizes and describes in a very detailed way CEA expression in cancer. It confirms and encurages to use CEA as serum marker for diagnostic and monitoring in cancer disease.
Figures and tables also in manuscript both in supplementary data are exaustive and of high quality.
References are representative of international scientific literature and updated.
I accept this manuscript for publish in this way.
Thank you
Reply:
We thank the Reviewer for the evaluation of our work.

Reviewer 2 Report
Comments and Suggestions for Authors
The authors present a comprehensive look at the expression of CEA on various tumor types using a standardized methodology, significantly reducing variables and uncertainties associated with reports on individual tumor type in the literature.
1. Can you comment on the comparison between baseline CEA expression in normal tissue types and the various tumor types you used?
2. Can you add the information on normal tissue expression to Table 1?
3. Why wasn't the same antibody used for the normal and tumor tissues?
4. Can you offer some discussion/speculation on CEA expression and adenocarcinomas versus other tumor types or origins? Could there be some explanation for the CEA expression variability?
Author Response
The authors present a comprehensive look at the expression of CEA on various tumor types using a standardized methodology, significantly reducing variables and uncertainties associated with reports on individual tumor type in the literature.
- Can you comment on the comparison between baseline CEA expression in normal tissue types and the various tumor types you used?
Reply: We have now added supplementary table 1 summarizing the details of CEA expression in normal tissues and commented on the often higher staining level of CEA in cancers as compared to the respective normal tissues on page 8, lines 186-188.
- Can you add the information on normal tissue expression to Table 1?
Reply: We have added a table with the CEA expression in normal tissue as supplementary table 1 and re-numbered the other supplementary tables accordingly.
- Why wasn't the same antibody used for the normal and tumor tissues?
Reply: The normal tissue was stained with the same antibody (MSVA-465R) as the tumor tissue. As mentioned in Mat&Methods (page 3, lines 119-123), the antibody (II-7) was only used for the purpose of validation. To avoid confusion, we have stated at the beginning paragraphs 3.2 (normal tissues) and 3.3 (cancer tissues) that the analysis was made with MSVA-465R.
- Can you offer some discussion/speculation on CEA expression and adenocarcinomas versus other tumor types or origins? Could there be some explanation for the CEA expression variability?
Reply: As a member of the family of epithelial CEACAMs, it is expected that CEA-positive tumors arise from epithelial and glandular tissues. We have now added this information to our discussion on page 13, line 237.

Reviewer 3 Report
Comments and Suggestions for Authors
The authors have used an interesting approach to evaluate CEA expression across a large number of tumor and normal tissue samples. Their use of consistent methodologies, including staining techniques, antibodies, and IHC, provides a comprehensive and standardized evaluation of CEA expression, which enhances the reliability of the findings. However, the title of the paper should be revised to reflect the actual number of tissue samples analyzed, as indicated in the results section. The study analyzed 13,725 tissue samples, not 15,413 as stated in the title. This discrepancy could be misleading. While the tables contain important information, their current layout makes them difficult to read and comprehend. A revision of the graphical presentation is necessary to improve clarity. The figures are excellent and convey the results effectively. However, Figure 3 could benefit from additional emphasis. The authors should consider adding critical commentary on the results shown in this figure, discussing their potential implications. The discussion is thorough and well-rounded. However, it would be valuable to further explore the role of CEA beyond its potential as a therapeutic target. The authors mention that CEA may not be directly related to the overall aggressiveness of tumors, which raises questions about how the high expression of CEA in certain epithelial tumors should be interpreted. How do the authors reconcile this with the existing literature or their findings? Additionally, while CEA-targeted therapies are mentioned as a challenging therapeutic path, it would be interesting to know how the findings of this study could contribute to future research or clinical applications, especially in light of the current limitations of targeting CEA.
Author Response
The authors have used an interesting approach to evaluate CEA expression across a large number of tumor and normal tissue samples. Their use of consistent methodologies, including staining techniques, antibodies, and IHC, provides a comprehensive and standardized evaluation of CEA expression, which enhances the reliability of the findings.
Reviewer: However, the title of the paper should be revised to reflect the actual number of tissue samples analyzed, as indicated in the results section. The study analyzed 13,725 tissue samples, not 15,413 as stated in the title. This discrepancy could be misleading.
Reply: We have revised the title as suggested by the reviewer.
Reviewer: While the tables contain important information, their current layout makes them difficult to read and comprehend. A revision of the graphical presentation is necessary to improve clarity. The figures are excellent and convey the results effectively.
Reply: We have adapted the layout of Table 1 and Table 2 for better readability.
Reviewer: However, Figure 3 could benefit from additional emphasis. The authors should consider adding critical commentary on the results shown in this figure, discussing their potential implications.
Reply: We have now commented of the data shown in Figure 3 on page 12, lines 226-227.
Reviewer: The discussion is thorough and well-rounded. However, it would be valuable to further explore the role of CEA beyond its potential as a therapeutic target.
Reply: We have extensively commented on the role of CEA as a diagnostic marker for the distinction of malignant mesothelioma from pulmonary adenocarcinoma and for the distinction of primary tumors from metastases in the liver (page 14, lines 262-275), its potential as a prognostic marker (page 14, lines 276-300), and as a serum marker (lines 301 – 319).
Reviewer: The authors mention that CEA may not be directly related to the overall aggressiveness of tumors, which raises questions about how the high expression of CEA in certain epithelial tumors should be interpreted. How do the authors reconcile this with the existing literature or their findings?
Reply: We have expanded our discussion on CEA upregulation on page 14, lines 286-287.
Reviewer: Additionally, while CEA-targeted therapies are mentioned as a challenging therapeutic path, it would be interesting to know how the findings of this study could contribute to future research or clinical applications, especially in light of the current limitations of targeting CEA.
Reply: We have now expanded the conclusions to highlight the importance of a comprehensive list of CEA-positive tumor types for potential future research.

Reviewer 4 Report
Comments and Suggestions for Authors
The manuscript "Carcinoembryonic antigen (CEA) expression in human tumors: A tissue microarray study on 15,413 tumors" reports on the comparative evaluation of the CEA expression in normal and neoplastic tissues by analyzed by highly standardized immunohistochemistry. This study identified CEA in 65 (54.2%) of 120 tumor categories, including 49 (40.8%) tumor types with at least one strongly positive case. The highest frequency of CEA expression was detected in colorectal adenomas and carcinomas, other gastrointestinal adenocarcinomas, medullary carcinomas of the thyroid, pulmonary adenocarcinoma, mucinous carcinomas of the ovary and the breast, small cell carcinomas of the lung, and urinary bladder. Moreover, CEA overexpression was linked to high tumor grade and invasive growth (p<0.0001 each) in urinary bladder cancer and estrogen and HER2 receptor positivity (p≤0.0158) in invasive breast cancer of no special type. In colorectal adenocarcinomas, reduced CEA expression was associated with mismatch repair deficiency (p<0.0001). This work identified CEA expression in a broad range of epithelial neoplasms; however, CEA serum monitoring might benefit for diagnostics but not for clinically relevant prognosis for most tumors.
These data may contribute to developing new cancer diagnostic and therapy approaches. The significance of the study is substantiated in the Introduction. The description of the methods and result presentation are accurate and sufficiently informative. The quality of the Figures and Tables is good. The Discussion section critically analyzes the results of previous studies. The list of references includes the relevant publications. In general, the presented results support the conclusions.
I have the following comment on the manuscript.
The Introduction, Conclusion, and Abstract should clearly state the novelty of the presented study.
Author Response
These data may contribute to developing new cancer diagnostic and therapy approaches. The significance of the study is substantiated in the Introduction. The description of the methods and result presentation are accurate and sufficiently informative. The quality of the Figures and Tables is good. The Discussion section critically analyzes the results of previous studies. The list of references includes the relevant publications. In general, the presented results support the conclusions.
I have the following comment on the manuscript.
The Introduction, Conclusion, and Abstract should clearly state the novelty of the presented study.
Reply: We have expanded the abstract (line 35), the introduction (page 2, lines 78-80), and the conclusions (lines 340-342) to highlight the usefulness of a comprehensive study of CEA expression under standardized conditions.
